# Fall-Related Hospitalizations in Elderly People: Temporal Trend and Spatial Distribution in Brazil

**DOI:** 10.3390/geriatrics8020030

**Published:** 2023-02-27

**Authors:** Glenda R. O. N. Ferreira, Tiago de N. das C. e Chagas, Lucia H. T. Gonçalves, Marília de F. V. de Oliveira, Eliã P. Botelho, Sandra H. I. Polaro

**Affiliations:** Programa de Pós-Graduação em Enfermagem, Universidade Federal do Pará, Belém 66075-110, PA, Brazil

**Keywords:** hospitalization, falls, elderly, temporal trend, spatial distribution, Brazil

## Abstract

This study aims to identify the temporal variation and the spatial dependence structure of the hospitalization rate for falls in the elderly residing in Brazil in the period between 2010 and 2021. This ecological study employs secondary data from the Brazilian Ministry of Health about the fall-related hospitalization of people aged 60 years old and over. A time-series analysis was carried out, employing the joinpoint model. For the spatial analysis, the Moran autocorrelation technique was employed. In Brazil, between 2010 and 2021, there were 1,270,341 hospitalizations for falls recorded among the elderly in the Brazilian Hospitalization System. There was a continuous upward trend between 2010 and 2019 for all age groups, female and male, and all Brazilian regions. The trend stabilized between 2019 to 2021. The North and Northeast regions had faster upward trends among all Brazilian regions, and there was also a faster upward trend among women compared to men. A high-high pattern in hospitalization incidence was noticed from 2011 to 2019 in the states of São Paulo, Minhas Gerais, Paraná, and Mato Grosso do Sul. The results of this study provide subsidies for Brazilian health authorities to implement more efficient public policies to improve the quality of life of elderly people.

## 1. Introduction

An important demographic and epidemiological transition has been observed over the past few decades in low- and middle-income countries, such as Brazil [1,2]. In high-income countries, an increase in the number of elderly people has been followed by an increase in the number of falls [3]. This is a multifactorial condition relating to demographic, economic, cultural, biological, psychological, cognitive, and social factors [4]. The severity of injuries in elderly people increases the risk of hospitalization, which leads to the higher use of emergency care services, higher occupancy of hospitals, and higher risks of death [3,4].

Biological factors contributing to falls among the elderly include knee osteoarthritis, low-back pain, diabetes mellitus, gait pattern variabilities, depression, dementia, stroke, Parkinson’s disease, rheumatic diseases, and urinary incontinence [5]. In addition, social and environmental factors, loneliness, social isolation, objects and pets, and a lack of non-slip mats in the kitchen and bedrooms and grab bars in the bathroom are examples of social and environmental factors associated with falls among the elderly [6,7,8].

In Brazil, people 60 years old and over are considered elderly [9]. The Brazilian Unified Health System (SUS) implemented public health policies as a response to the process of demographic and epidemiological transitions. The Elderly-Specific National Health Policy was approved in 2006 [10]. The network for the care of people with chronic diseases in the context of the Unified Health System was redefined in 2014 [11]. Despite this progress, Brazil is marked by geographic inequalities present in this service network, which affects the socioeconomic conditions of elderly people [12,13,14]. Furthermore, socioeconomic differences can influence the aging process, with adverse health consequences [15].

Recent studies on fall-related hospitalizations in elderly people have focused only on temporal trends [16,17], without identifying spatial areas with higher fall-related hospitalization rates. While a temporal study permits the evaluation of the influence of public policy on a specific problem, the spatial analysis shows the greater problematic areas that the health authorities should prioritize.

Therefore, the main goal of this study is to temporally and spatially analyze the fall-related hospitalization rate of elderly people living in Brazil between 2010 and 2021.

## 2. Materials and Methods

### 2.1. Study Design

This is an observational ecological study employing secondary data from the Brazilian Ministry of Health from 2010 to 2021.

### 2.2. Settings

The analysis units were Brazil as a whole, its macro-regions, and its states. The data were collected in September and October 2021 and February 2022. The Brazilian territory is divided into five regions, namely the North or Brazilian Amazon region, Northeast, Midwest, Southeast, and South. Brazil is ranked 84th in terms of the HDI (0.765) worldwide. The HDI disparity can also be noticed in Brazil, in which the North region, notably, has the lowest HDI (0.719) [12]. The Southeast region has the lowest coverage of the Primary Healthcare Network (50.99%) [13]. Additionally, access to the Brazilian public healthcare system varies among regions, with the North and Northeast regions having the worst access [14]. Most of the Brazilian elderlies live in urban zones and are household heads, with the Southeast and Northeast regions concentrating the greatest portion of the elderly population [18].

### 2.3. Participants

In this study. we included any notifications of hospitalizations caused by falling among people 60 years old and over in Brazil. All data were double-checked, and all inconsistencies were fixed.

### 2.4. Data Sources

The notification of hospitalizations by falling was obtained in DATASUS. DATASUS is the informatic system of SUS, responsible for collecting, processing, and disseminating health information in Brazil.

The population data were obtained from the Brazilian Institute of Geography and Statistics (IBGE).

### 2.5. Variables

We collected the following variables in DATASUS: age, gender (female and male), state and region of residency, and year of notification.

The annual fall-related hospitalization rates were calculated adopting the Brazilian Ministry of Health methods. The number of notifications was found by the projected elderly population for the specific region and state. The results were then multiplied by 100,000. For 2021, data for December were not available. Therefore, averages were calculated for the period between January and November.

### 2.6. Data Analysis

Temporal analyses were carried out, considering the annual incidences of fall-related hospitalizations for the age groups (60–64, 65–69, 70–74, 75–79, and 80 years old and over), genders, and Brazilian regions. The fall-related hospitalization rate was considered the dependent variable and the years 2010 to 2021 was the independent one. To eliminate the influence of population variation on the fall-related hospitalization rate calculus, we adjusted the rate by age groups employing direct methods.

The temporal analysis was carried out by employing the Joinpoint^®^ software (version number 4.8.0.1—National Cancer Institute, Calverton, MD, USA) and its segmented linear regression model was applied. The joinpoint method evaluates whether, at some points, there are alterations in the observed trend patterns. For this type of analysis, the software obtains trend data and fits the simplest possible joinpoint model, beginning with the minimum number of joinpoints and testing whether more joinpoints are statistically significant to be added to the model. For each trend segment, the software calculates the annual percent change (APC) that assumes that rates change at a constant percentage of the previous year’s rate.

The segmented linear regression model uses statistical criteria to determine when and how often the APC changes. The average annual percent change (AAPC) is a summarized trend measure that describes the average APC over a period of several years with a single number. It is a weighted average of the joinpoint model’s APC, with weights equal to the APC interval. The final step of the calculation converts the weighted average of slope coefficients into an annual percentage change.

For the spatial visualization of annual fall-related hospitalization rates in elderly people in Brazilian states, the calculated rates were adjusted by applying a local empirical Bayes method that considers only information about the rates of the neighbor states. The exploratory visualization of spatial data applied Moran’s statistics, since this allows the identification of the presence or absence of spatial autocorrelation of the variable fall-related hospitalization rate in elderly people between the states each year. For the years with significant results, the detection and presentation of distribution patterns of the rates as clusters followed. To smooth the fluctuation associated with small areas, a local empirical Bayes estimator was calculated. This is a weighted average of the local average rate that includes only geographic neighbors of the area for which the rate must be estimated. Subsequently, a contiguity matrix was obtained. The next step was carrying out Moran’s global univariate analysis and applying the method of the local indicators of spatial association by using the local Moran statistics. The data were presented by using the Moran map and showed the spatial representation of the clusters. A level of statistical significance at *p* < 0.05 was adopted [19,20].

Data were georeferenced and analyzed by using Terra View^®^ geographic information system software, version 4.2.2. The georeferenced meshes in a shapefile (.shp) format of state borders in Brazil were obtained from the Brazilian Institute of Geography and Statistics and projected in the longlat projection system, Datum Horizontal SIRGAS-2000. The results are shown in choropleth maps.

Descriptive statistics, including measures of central tendency and dispersion, as well as box plots and time-series graphs, were obtained by using the Minitab 20^®^ software.

The researchers did not have contact with the people who were interned since secondary data sources in the public domain were used, which were exempt from appreciation by the Research Ethics Committee, as recommended by the sole paragraph of article 1 of resolution nº 510, of 7 April 2016, from the National Health Council. The researchers respected all ethical guidelines for research with human beings, as recommended by CNS Resolution No. 466 of 12 December 2012.

## 3. Results

In Brazil, 1,270,341 fall-related hospitalizations were reported between 2010 and 2021. In this population, the age-adjusted rate in 2010 was 39.7 hospitalizations per 10,000 people (77,857). In 2021, the rate increased to 43.3 hospitalizations (131,970). In all of the analyzed periods, the North and Northeast Regions had lower rates than those obtained for the country, whereas the other regions had higher rates. The rate calculated for the Center-West region was lower than the national rate only in 2021 (Figure 1A). The box plot in Figure 1B shows the differences between the average rates of the regions and the national rate, which was 42.6 hospitalizations per 10,000 people (SD = 2.23, 95% CI: 41.2; 44.1).

The segmented linear regression model allowed us to understand and analyze the annual changes in the studied indicator. The fall-related hospitalization rate in elderly people showed a significant upward trend from 2010 to 2019 in Brazil when data were disaggregated by gender, age group, and region. However, there was a nonsignificant reduction from 2019 onwards. The only exception was the North region, whose change period was between 2015 and 2021, although statistical significance was observed in none of the periods (Table 1).

From 2010 to 2019, there was a significant growth trend in Brazil in the age-adjusted fall-related hospitalization rate in elderly people, with an annual percentage increase of 1.8% (95% CI: 1.3% to 2.3%; *p* = 0.00). This growth percentage was higher only in the Northeast and Center-West regions, with 4.7% (95% CI: 3.7; 5.8; *p* = 0.00) and 2.1% (95% CI: 0.4; 3.9; *p* = 0.00), respectively. The annual increase percentage for the female gender was 2.0 (95% CI: 1.5; 2.6; *p* = 0.00), higher than the rate for the male gender. Regarding age group, despite the high rates in elderly people 80 years old or older, the highest APC in all age groups, 2.0% (95% CI: 1.3; 2.8; *p* = 0.00), was found for elderly people from 60 to 64 years old (Table 1).

Descriptive analysis of the gross fall-related hospitalization rate in elderly people in the states, as shown in the choropleth maps, indicated that over the 12 years of the time series only Amazonas maintained rates lower than 4.7 hospitalizations per 10,000 people, whereas Minas Gerais, São Paulo, and Santa Catarina had rates higher than 42.7 hospitalizations per 10,000 people. From 2017 onwards, all the states in the South region showed the highest rates (Figure 2).

Exploratory visualization by means of Moran’s global method found that there was no spatial autocorrelation in the smoothed fall-related hospitalization rate in elderly people in 2009 (GMI = 0.17; *p* = 0.12), 2010 (GMI = 0.23; *p* = 0.06), 2014 (GMI = 0.07; *p* = 0.27), 2015 (GMI = 0.09; *p* = 0.24), 2016 (GMI = 0.15; *p* = 0.12), 2017 (GMI = 0.07; *p* = 0.30), and 2018 (GMI = 0.13; *p* = 0.14). However, the studied indicator showed spatial autocorrelation in 2011 (GMI = 0.26; *p* = 0.04), 2012 (GMI = 0.39; *p* = 0.01), 2013 (GMI = 0.29; *p* = 0.02), 2019 (GMI = 0.30; *p* = 0.02), 2020 (GMI = 0.28; *p* = 0.03), and 2021 (GMI = 0.29; *p* = 0.02).

In the years in which spatial autocorrelation was found in the studied indicator, Moran’s local methodology was applied to identify area clusters. From 2011 to 2013 and in 2019, there was only one spatial pattern: states with high rates of fall-related hospitalizations in elderly people correlated with states in the same situation. In 2011 and 2019, a single cluster with a high-high pattern combined the states of Mato Grosso do Sul, Paraná, São Paulo, and Minas Gerais. In 2012, the Federal District became a part of this cluster. In 2013, Paraná left it. This cluster and this pattern were no longer observed only from 2020 onwards, when a high-low pattern began to exist in the North region (Roraima, Amazonas, Acre, and Rondônia) (Figure 3).

## 4. Discussion

Our results showed higher Brazilian elderly fall-related hospitalization rates. Among the Brazilian regions, the North and Northeast regions had the lowest rates. Temporal analysis revealed two trend periods: an upward trend (2010–2015), followed by a stabilized one (2015–2021). In the first trend period, both genders, male and female, and all age range groups had an upward trend. The North and Northeast regions had the highest upward trend of elderly fall-related hospitalization rates. Spatial analysis indicated a high-high cluster of fall-related hospitalization between 2011 and 2019, comprising the states of São Paulo, Minas Gerais, Paraná, and Mato Grosso do Sul.

Although we noticed a stabilized trend in hospitalizations related to falls among the elderly between 2019 and 2021, this change may be due the lockdown imposed by the COVID-19 pandemic. Although the AAPC showed an upward trend for fall-related hospitalizations only for females and the 60–64 age group, this could be due to the balance promoted by the APC of the second trend period.

The higher upward trend in fall-related hospitalization rates noticed in the North and Northeast Brazilian regions could be due to the worse quality of life in these regions. Among all Brazilian regions, the North and Northeast are the poorest regions with the lowest access to healthcare services [13,14]. A study among British elderly people showed that those living at a lower socioeconomic status had an accelerated aging process and greater adverse changes in physical capabilities [15].

Our results also showed that fall-related hospitalization rates increased faster in women than in men. In Alagoas, Brazil, a time-series study from 2008 to 2019 also showed the same result [21]. A recent study among elderly people also showed a higher risk of falls among women associated with the disruption of gait patterns [22].

Furthermore, the widowhood process strongly influences the health of elderly women; this is due both to the challenge of dealing with loneliness due to the absence of a spouse, and to the issue of solitary management of the home itself, which does not always find favorable conditions [23,24]. Such situations increase the chances of depression, hospitalization in long-term institutions, the development of dementia, and the occurrence of falls [25].

Our results showed a stable high-high cluster comprising the states of São Paulo, Mato Grosso do Sul, Paraná, and Minas Gerais from 2011 to 2019. São Paulo, Minas Gerais, and Paraná are the first, second, and fifth highest cities with the greater Brazilian population. In addition, a time-series study between 1996 and 2012 showed that in the Southeast region, the capitals of São Paulo and Minas Gerais had a greater APC of fall-related hospitalization among the elderly than the other capitals of the Southeast regions [26].

Brazilian Law 10,471 establishes that elderly people should enjoy all fundamental rights inherent to a human being and they must be assured physical and mental healthcare [10]. However, the continuous upward trend of fall-related hospitalization among Brazilian elderlies, as well as the stable high-high hospitalization rate cluster shown in our study, suggest that these rights have not been guaranteed by Brazilian authorities. The low efficacies of public policies promoting the quality of life in elderlies are worrying when considering the aging population. It is even more worrying considering that by the end of 2050, about 80% of older people will be living in developing countries, such as Brazil [27].

From 2000 to 2020, Brazil spent USD 684,277,778 on the fall-related hospitalization of elderly people [28]. Promoting quality of life to the elderly would save the Brazilian budget and would permit the country to invest in other necessary areas.

The main limitation of the present study was the use of data collected from information systems, which involves the risk of using incorrect or incomplete information. In addition, as an ecological study, we cannot discuss the casual effects. Other studies must be carried out focusing on specific possible factors contributing to falls among Brazilian elderly people. Other studies must be also carried out in order to study the effect of the COVID-19 pandemic on the fall-related hospitalization of the elderly.

## 5. Conclusions

There was an upward trend in fall-related hospitalizations in all age groups of elderly people and in both females and males in Brazil from 2010 to 2019. The results indicated different regional patterns in the upward trends among Brazilian regions, with the North and Northeast regions having the greatest upward trends. The stabilization in the trends noticed between 2019 to 2021 may be due to the COVID-19 pandemic. The spatial analysis pointed to a stable high-high cluster comprising the states of São Paulo, Minas Gerais, Paraná, and Mato Grosso do Sul.

Rather than only structuring the healthcare services to provide better assistance to elderlies and their families, the Brazilian municipal, state, and federal authorities should focus on improving the quality of life of Brazilians through an inclusive policy that respects each person or age group.

## Figures and Tables

**Figure 1 geriatrics-08-00030-f001:**
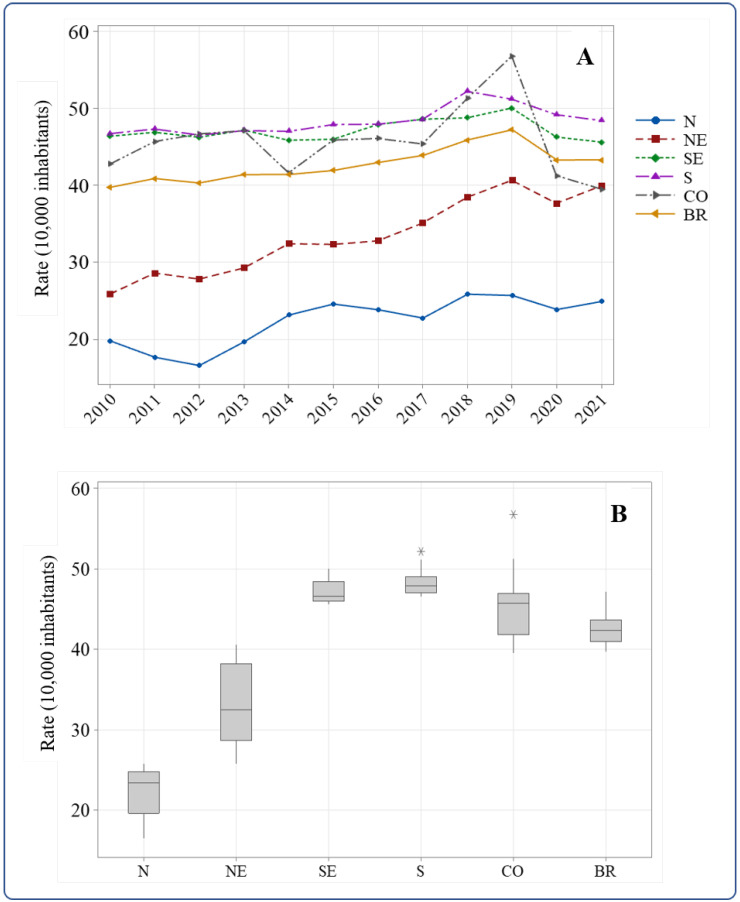
Temporal distribution (**A**) and box plot for median and quartiles (**B**) of the age-adjusted fall-related hospitalization rate in elderly people in Brazil and its regions, 2010 to 2021. * Outlier. Regions: N—North; NE—Northeast; SE—Southeast; S—South; CO—Center-West; BR—Brazil. Source: Brazil [13].

**Figure 2 geriatrics-08-00030-f002:**
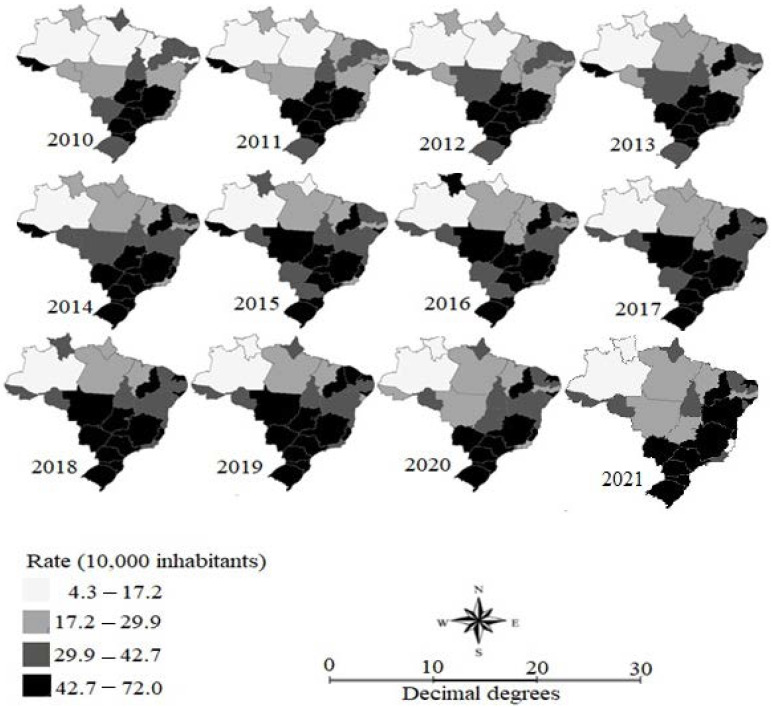
Thematic mapping of the smoothed fall-related hospitalization rate in elderly people per 10,000 people in Brazilian federative units, 2010 to 2021. Source: Brazil [13].

**Figure 3 geriatrics-08-00030-f003:**
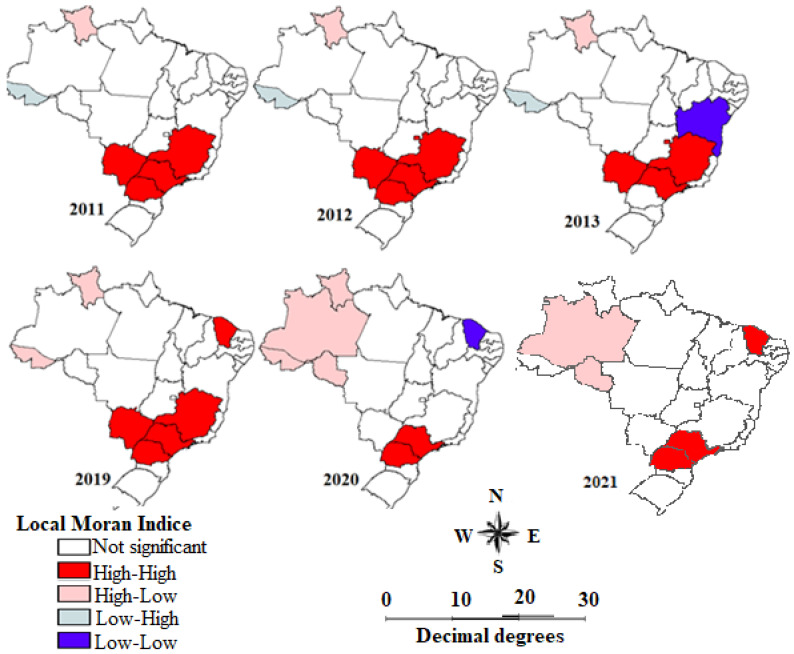
Exploratory visualization mapping of the smoothed fall-related hospitalization rate in elderly people by Brazilian federative unit, 2010 to 2021, using Moran’s map. Source: Brazil [13].

**Table 1 geriatrics-08-00030-t001:** Percentage change in the age-adjusted and gross fall-related hospitalization rate in elderly people (per 10,000 people). Overall data for Brazil and disaggregated data by gender, age groups, and region, 2010 to 2021.

Variable	Rate		APC ^b^ (95% CI ^c^)		AAPC ^e^ (2010 to 2021)
	2010	2021	2010 to 2019	2019 to 2021	(95% CI ^c^)
Brazil	39.7	43.3	1.8(1.3; 2.3) ^d^	−3.5(−8.1; 1.3)	0.8(−0.0; 1.6)
Gender					
Female	42.7	47.7	2.0(1.5; 2.6) ^d^	−3.8(−8.5; 1.1)	0.9(0.1; 1.8) ^d^
Male	35.9	37.8	1.4(0.9; 2.0) ^d^	−3.0(−7.8; 2.0)	0.6(−0.2; 1.5)
Regions					
Northeast	25.8	39.9	4.7(3.7; 5.8) ^d^	−0.2(−8.7; 9.2)	3.8(2.2; 5.4) ^d^
Southeast	46.4	45.6	0.8(0.2; 1.3) ^d^	−3.9(−8.9; 1.4)	−0.1(−1.0; 0.8)
South	46.7	48.4	1.2(0.4; 1.9) ^d^	−2.5(−9.3; 4.9)	0.5(−0.7; 1.7)
Center-West	42.7	39.5	2.1(0.4; 3.9) ^d^	−13.9(−27.2; 1.9)	−1.0(−3.7; 1.8)
North	19.7	25.9	6.5 (−0.5; 14.1)	1.0(−3.1; 5.3)	2.2(−3.7; 8.5)
Age groups (years) ^a^					
60 to 64	24.5	27.6	2.0(1.3; 2.8) ^d^	−2.0(−8.7; 5.1)	1.3(0.1; 2.4) ^d^
65 to 69	28.8	31.4	1.7(0.9; 2.5) ^d^	−2.7(−9.5; 4.6)	0.9(−0.3; 2.1)
70 to 74	37.1	39.5	1.4(0.9; 1.8) ^d^	−3.1(−7.2; 1.1)	0.5(−0.2; 1.3)
75 to 79	50.7	54.8	1.8(1.2; 2.5) ^d^	−5.2(−10.7; 0.6)	0.5(−0.5; 1.5)
80 or more	88.4	94.6	1.8(1.4; 2.2) ^d^	−4.8(−8.5; −1.0)	0.6(−0.1; 1.2)

^a^ Gross rate; ^b^ annual percentage change; ^c^ confidence interval; ^d^
*p*-value < 0.05; ^e^ average annual percentage change. Source: Brazil [13].

## Data Availability

Not applicable.

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
