# Peer review of "Fall-Related Hospitalizations in Elderly People: Temporal Trend and Spatial Distribution in Brazil"

_geriatrics, 2023, doi:10.3390/geriatrics8020030_

Round 1

Reviewer 1 Report

The manuscript “Fall-related hospitalizations in elderly people: Temporal trend 2 and spatial distribution in Brazil” investigate an important topic, which is common and causing much suffering for older people. The study is based on big data sets, but the text need to be clarified and more structured. 

Introduction
It would be of interest for the reader to receive information about what kind injuries/diagnoses are related to older peoples falls.

You write about secondary diagnoses, and it is not clear what it means.

On line 38 and 39 it describes that hospital and death rate varies by gender and age groups. In what way?

Objective/aim
The objective differs, in the abstract there is one objective and in text prior “Materials and Methods” there are two other objectives. They should be the same.

In the text prior “Materials and Methods” there is a mix of research question and objectives. I would be clearer if you first present a distinctly objective and then a related research question.

Materials and Methods
The text in this section is difficult to follow, as it is a mix of design, sample, data collection and analysis. I suggest a use of subheadings: 2.1 Design, 2.2 Sample or Participants, 2.3 Data collection and 2.4 Analysis.

Design should be shortly described in one sentence.

Sample/Participants should describe the participants included in the study, criterions for inclusion/exclusion and recruitment.

Data collection should describe how the data were collected, which variables, from which sources and the procedure.

Analysis should clearly describe how the data were analysed. Some clarifications are needed.

On line 90: why are you multiplying by 10,000? What is the “direct method” on line 91?

On line 104 I suppose you refer to the “linear regression model” on line 96?

Do you have references to “Bayes method” and “Moran´s statistics”?

It is unclear if the study was assessed by a research ethics committee or not. If so, is there a diary number?

Results

The results are mixed up with some information about the setting, data collection and analysis, which should be removed or transferred (if applicable) to the section “Materials and Methods”.

In figure 1 it is difficult to read off the line for the regions in black/white outprint, respectively. Could you please make it clearer to observe the various lines.

You describe “In 2021, the rate increased to 43.3 hospitalizations (131,970). In all of the analyzed period, the North and Northeast Regions had lower rates than those obtained for the country, whereas the other regions had higher rates. The rate calculated for the Center-West region was lower than the national rate only in 2021 (Figure 1A)”. Did you calculate any p-level for the increase, lower rates and higher rates, respectively?

Related to Table 1 you expressed that “The fall-related hospitalization rate in elderly people showed a significant upward trend ….”, but I can only observe 3 significant differences regarding Female, Region: Northeast and Age group:60 to 64. Could you please clarify?

In the text after Table 1a number of significant results are presented, but dp-values <0.05 cannot be found in the table. Please clarify.

Discussion

You are reasoning about feminization of old age related to fall-related hospitalization rates are interesting. Perhaps you in addition might include the social factor that women often have to manage by themselves as widower in last part of life, while men often receive support from their wife.

Author Response

Dear reviewer,

We are thankful for your suggestions since they brought more quality to the paper. We attended all of them and below you can see all responses.

Thank you once again!

REVIEWER 1:
The manuscript “Fall-related hospitalizations in elderly people: Temporal trend 2 and spatial distribution in Brazil” investigate an important topic, which is common and causing much suffering for older people. The study is based on big data sets, but the text need to be clarified and more structured.  

Introduction
It would be of interest for the reader to receive information about what kind injuries/diagnoses are related to older peoples falls.

RESPONSE: On page 1 (lines 32-39) we include this information now “Biological factors contributing to falls among the elderly include knee osteoarthritis, low-back pain, diabetes mellitus, gait pattern variabilities, depression, dementia, stroke, Parkinson's disease, rheumatic diseases, and urinary incontinence [5]. In addition, social and environmental factors, loneliness, social isolation, objects and pets, non-slip mats in the kitchen and bedrooms, and a lack of grab bars in the bathroom are examples of social and environmental factors associated with falls among the elderly [6,7,8,9].”  

You write about secondary diagnoses, and it is not clear what it means.

Response: We removed this term now

On line 38 and 39 it describes that hospital and death rate varies by gender and age groups. In what way?

Response: We removed it now.

Objective/aim
The objective differs, in the abstract there is one objective and in text prior “Materials and Methods” there are two other objectives. They should be the same.

Response: We removed it from the methods since there is no point to repeat the main goal of the study in methods section

Materials and Methods
Response: Thank you for the suggestion! We did it!

Analysis should clearly describe how the data were analysed. Some clarifications are needed. 

On line 90: why are you multiplying by 10,000? What is the “direct method” on line 91? 

Response: We multiplied by 10,000 because we followed the same methods of the Ministry of Health. Therefore, our results could be compared with theirs. We included this information on page 2, lines 83-84.

Direct method refers to the calculation method used to standardize the rate. It removed the influences of population variation on the calculus of the fall-related hospitalization rates by uniformizing the intra-age variations. We included this information on page 2, lines 93-95.

“To eliminate the influence of population variation on the fall-related hospitalization rate calculus, we adjusted the rate by age groups employing direct methods”.

On line 104 I suppose you refer to the “linear regression model” on line 96? 

Response: Include. Page 3, line 97.

Do you have references to “Bayes method” and “Moran´s statistics”? 

Response: Included refs. 18 and 19 (page 3, line 124).

It is unclear if the study was assessed by a research ethics committee or not. If so, is there a diary number?

Response: we included this information now on page 3, lines 132-137. “The researchers did not have contact with the people who were interned since secondary data sources in the public domain were used, which were exempt from appreciation by the Research Ethics Committee, as recommended by the sole paragraph of article 1 of resolution nº 510, of 7th April 2016, from the National Health Council. The researchers respected all ethical guidelines for research with human beings, as recommended by CNS Resolution No. 466 of 12th December 2012.”

Results

The results are mixed up with some information about the setting, data collection and analysis, which should be removed or transferred (if applicable) to the section “Materials and Methods”. 

Response: All the results were rewritten.

In figure 1 it is difficult to read off the line for the regions in black/white outprint, respectively. Could you please make it clearer to observe the various lines.

Response: We changed now for colors. It is clearer now. Thank you!

You describe “In 2021, the rate increased to 43.3 hospitalizations (131,970). In all of the analyzed period, the North and Northeast Regions had lower rates than those obtained for the country, whereas the other regions had higher rates. The rate calculated for the Center-West region was lower than the national rate only in 2021 (Figure 1A)”. Did you calculate any p-level for the increase, lower rates and higher rates, respectively?

Response: No, we don’t. Here we just described how was the rates for the regions and Brazil. We did not apply any statical analysis.  

Related to Table 1 you expressed that “The fall-related hospitalization rate in elderly people showed a significant upward trend ….”, but I can only observe 3 significant differences regarding Female, Region: Northeast and Age group:60 to 64. Could you please clarify?

In the text after Table 1a number of significant results are presented, but dp-values <0.05 cannot be found in the table. Please clarify.

Response: We corrected it now.

Discussion

You are reasoning about feminization of old age related to fall-related hospitalization rates are interesting. Perhaps you in addition might include the social factor that women often have to manage by themselves as widower in last part of life, while men often receive support from their wife.

 Response: All discussion was rewritten and we inserted this information on lines 229 to 237 (page 7 and 8).

Reviewer 2 Report

This research focuses on a very interesting study field, fall-related hospitalizations in elderly people. However, the manuscript has several weaknesses, in the study support, methods, results, discussion, and conclusions.

Major comments

11. Introduction 

The authors must include the scientific support that justifies their study. In this sense, they have to highlight the novelty of their  study. They note that "the present study proposed the following research question: Are there differences in the temporal variations and structure of spatial dependence of the fall-related hospitalization rate for elderly people living in Brazil between 2010 and 2021?". The answer is surely yes, but what is really relevant would be why?

2 2.Method

--The authors must specify the differential characteristics of the studied regions regarding the socioeconomic level of the population, environmental characteristics, health services and possible social support networks (volunteering), with whom they live (they are alone at home).

      3.Results

--The results are too much limited. In this sense, it is necessary for the authors to present the data stratified by age and sex in the regions, in order to identify the risk.

--It is also necessary that they consider other factors such as  education   by  region area, socioeconomic level, how many people live at home (percentage of people who live alone) and other possible factors associated with the risk of falls.

4. Discussion.

-The discussion needs to be rewritten.

- Authors should highlight the relevance of their findings and their implications,  considering the differential characteristics of the regions regarding the variables related to schooling, sex, economic income, comorbidity, basic and instrumental activities of daily life, social support networks, as well as type and available health services.

5. Conclusions

- The authors should include some proposal regarding public policies or health services,  as an alternative to the epidemiological problem identified in the study,  regarding fall-related hospitalizations in elderly people in Brazil and its possible extrapolation to countries with similar characteristics.

Author Response

Dear reviewer,

We are thankful for your suggestions since they brought more quality to the paper. We attended all of them and below you can see all responses.

Thank you once again!

REVIEWER 2:
This research focuses on a very interesting study field, fall-related hospitalizations in elderly people. However, the manuscript has several weaknesses, in the study support, methods, results, discussion, and conclusions.

Major comments

  1. Introduction

The authors must include the scientific support that justifies their study. In this sense, they have to highlight the novelty of their  study. They note that "the present study proposed the following research question: Are there differences in the temporal variations and structure of spatial dependence of the fall-related hospitalization rate for elderly people living in Brazil between 2010 and 2021?". The answer is surely yes, but what is really relevant would be why?

Response: On pages 1 and 2 (lines 43 to 51) we clarified the socioeconomic differences of Brazilian regions and let clear that no spatiall analysis has been done yet. We also showed how the temporal and spatial analyses are important to health authorities.

2 2.Method

--The authors must specify the differential characteristics of the studied regions regarding the socioeconomic level of the population, environmental characteristics, health services and possible social support networks (volunteering), with whom they live (they are alone at home).

Response: We did it on page 2 (lines 59-68). “The analysis units were Brazil as a whole, its macro-regions, and its states. The data were collected in September and October 2021 and February 2022. The Brazilian territory is divided into five regions, namely, the North or Brazilian Amazon region, Northeast, Midwest, Southeast, and South. Brazil is ranked 84th in terms of the HDI (0.765) worldwide. The HDI disparity can also be noticed in Brazil, in which the North region, notably, has the lowest HDI (0.719) [13]. The Southeast region has the lowest coverage of the Primary Healthcare Network (50.99%) [14]. Additionally, access to the Brazilian public healthcare system varies among regions, with the North and Northeast regions having the worse access [15]. Most of the Brazlian elderlies live in urban zones, are househeads, with the Southeast and Northeast regions concentrating the greatest portion of elderlies population [19]. “

      3.Results

--The results are too much limited. In this sense, it is necessary for the authors to present the data stratified by age and sex in the regions, in order to identify the risk.

--It is also necessary that they consider other factors such as  education   by  region area, socioeconomic level, how many people live at home (percentage of people who live alone) and other possible factors associated with the risk of falls.

Response: We understand your suggestion, but this is part of another study in progress by our group. First we decide to show a panoramic reality of the studied phenomenon in Brazil. In the study in progress we are employing spatial regression to see which social determinants of health are implicated in the problem and also  and spatial scan to measure the spatio-temporal risks

4  4. Discussion. 

-The discussion needs to be rewritten.

- Authors should highlight the relevance of their findings and their implications,  considering the differential characteristics of the regions regarding the variables related to schooling, sex, economic income, comorbidity, basic and instrumental activities of daily life, social support networks, as well as type and available health services.

Response: All this topic was rewritten. Discussion is more concise and focusing on our results.

  1. Conclusions

- The authors should include some proposal regarding public policies or health services,  as an alternative to the epidemiological problem identified in the study,  regarding fall-related hospitalizations in elderly people in Brazil and its possible extrapolation to countries with similar characteristics.

Response: We rewritten the conclusion include the following sentence: “Rather than only structuring the healthcare services to provide better assistance to elderlies and their families, the Brazilian municipal, state, and federal authorities should focus on improving the quality of life of Brazilians through an inclusive policy that respects each person or age group.” (page 8, lines 274-277).

Round 2

Reviewer 2 Report

The authors have corrected the manuscript considering all the comments.

The manuscript substantially improved, in the theoretical support, method, results and discussion.